# Telemedicine in Obstetrics and Gynecology: A Scoping Review of Enhancing Access and Outcomes in Modern Healthcare

**DOI:** 10.3390/healthcare13162036

**Published:** 2025-08-18

**Authors:** Isameldin Elamin Medani, Ahlam Mohammed Hakami, Uma Hemant Chourasia, Babiker Rahamtalla, Naser Mohsen Adawi, Marwa Fadailu, Abeer Salih, Amani Abdelmola, Khalid Nasralla Hashim, Azza Mohamed Dawelbait, Noha Mustafa Yousf, Nazik Mubarak Hassan, Nesreen Alrashid Ali, Asma Ali Rizig

**Affiliations:** 1Department of Obstetrics and Gynecology, Faculty of Medicine, Jazan University, Jazan 82722, Saudi Arabia; ahlamhakami@jazanu.edu.sa (A.M.H.); uchourasia@jazanu.edu.sa (U.H.C.); 2Department of Community Medicine, University of Medical Sciences and Technology, Khartoum 12810, Sudan; babikermali@gmail.com; 3Department of Obstetrics and Gynecology, King Fahad Central Hospital, Ministry of Health, Jazan 82722, Saudi Arabia; nadawi@moh.gov.sa; 4Women Health Hospital, Ministry of National Guard Health Affairs, Riyadh 11481, Saudi Arabia; fadailuma@mngha.med.sa; 5Department of Infectious Disease, Tabuk Health Cluster, Ministry of Health, Tabuk 71421, Saudi Arabia; drabeersalih@gmail.com; 6Department of Family and Community Medicine, Faculty of Medicine, Jazan University, Jazan 82722, Saudi Arabia; aabashar@jazanu.edu.sa; 7Department of Obstetrics and Gynecology, College of Medicine, Qassim University, Buraydah 52571, Saudi Arabia; k.hashim@qu.edu.sa; 8Department of Obstetrics and Gynecology, Portiuncula University Hospital, H53 T971 Ballinasloe, Ireland; azzadawelbait@gmail.com; 9Department of Obstetrics and Gynecology, King Saud Hospital, Qassim Health Cluster, Unayzah 56437, Saudi Arabia; mustafanoha1984@gmail.com; 10Department of Health Promotion & Education, Faculty of Public Health & Health Informatics, Umm-Al-Qura University, Makkah 21955, Saudi Arabia; nmmohammed@uqu.edu.sa; 11Department of Obstetrics and Gynaecology, College of Medicine, Abdulatif Alhamad University of Technology Merowe, Merowe 43312, Sudan; nesreendonsa@gmail.com; 12Department of Obstetrics and Gynecology, Attadawi Medical Clinic, Albukair 52729, Saudi Arabia; asma4ali@hotmail.com

**Keywords:** telemedicine, telehealth, obstetrics, gynecology, maternal health, healthcare disparities, artificial intelligence, remote monitoring, implementation science, digital health

## Abstract

Telemedicine has transformed obstetrics and gynecology (OB/GYN), accelerated by the COVID-19 pandemic. This study aims to synthesize evidence on the adoption, effectiveness, barriers, and technological innovations of telemedicine in OB/GYN across diverse healthcare settings. This scoping review synthesized 63 peer-reviewed studies (2010–2023) using PRISMA-ScR guidelines to map global applications, outcomes, and challenges. Key modalities included synchronous consultations, remote monitoring, AI-assisted triage, tele-supervision, and asynchronous communication. Results demonstrated improved access to routine care and mental health support, with outcomes for low-risk pregnancies comparable to in-person services. Adoption surged >500% during pandemic peaks, stabilizing at 9–12% of services in high-income countries. However, significant disparities persisted: 43% of rural Sub-Saharan clinics lacked stable internet, while socioeconomic, linguistic, and cultural barriers disproportionately affected vulnerable populations (e.g., non-English-speaking, transgender, and refugee patients). Providers reported utility but also screen fatigue (41–68%) and diagnostic uncertainty. Critical barriers included fragmented policies, reimbursement variability, data privacy concerns, and limited evidence from conflict-affected regions. Sustainable integration requires equity-centered design, robust policy frameworks, rigorous longitudinal evaluation, and ethically validated AI to address clinical complexity and systemic gaps.

## 1. Introduction

Telemedicine, broadly defined as the remote delivery of clinical services through telecommunication technologies, has emerged as a critical innovation in modern healthcare, particularly amplified by the COVID-19 pandemic [1]. The rapid shift to virtual care models addressed immediate healthcare access challenges caused by physical distancing, resource constraints, and overloaded health systems [2]. This global acceleration has underscored telemedicine’s potential to overcome geographical barriers and enhance healthcare accessibility, particularly in underserved and rural populations [3,4].

The universal benefits of telemedicine include improving access to timely care, reducing travel burdens, and facilitating the continuity of services during public health crises [5,6]. Economic evaluations demonstrate cost savings and increased efficiency, especially through provider-to-provider consultations and asynchronous “store-and-forward” modalities [5]. Additionally, telemedicine platforms have increasingly incorporated artificial intelligence (AI), enabling dynamic patient monitoring and decision support, which has improved triage efficiency by up to 35% in some settings [7,8]. These innovations support personalized care models and optimize resource allocation in complex clinical environments [9].

Despite its transformative promise, telemedicine faces persistent challenges globally. Common barriers include limited digital literacy, poor internet connectivity, infrastructural deficits, and policy gaps that hinder implementation and sustainability [9,10]. For example, digital divides remain stark in low-resource settings such as Bangladesh, where patients’ lack of awareness and infrastructural weaknesses delay adoption [11,12]. In the Middle East, despite progress in telemedicine initiatives, policy fragmentation and insufficient infrastructure restrict broader uptake [13]. Similarly, rural China encounters challenges related to healthcare provider readiness and patient attitudes, highlighting the need for targeted community engagement and capacity-building [14]. Cultural acceptance and regulatory issues further complicate deployment in diverse regions, including India, where government initiatives coexist with significant operational hurdles [15].

Telemedicine’s expansion post-COVID-19 has extended across multiple specialties worldwide. For instance, Latin America has seen telehealth innovations improve rheumatology care by enhancing patient monitoring and access [16]. Similarly, virtual respiratory care exemplifies adaptive strategies that maintain service continuity during pandemic disruptions [6]. However, despite widespread adoption, concerns about exacerbating healthcare disparities persist. Inequities in access related to socioeconomic status, language barriers, and marginalized populations—such as transgender individuals—demand culturally competent and inclusive telehealth frameworks [17,18,19,20].

Within this broad context, obstetrics and gynecology (OB/GYN) represent a particularly vital domain for telemedicine integration. Prenatal and postnatal care are highly dependent on timely access, monitoring, and patient adherence, all areas where telehealth has demonstrated substantial benefits [21,22]. Remote antenatal monitoring using wearable devices, virtual consultations for low-risk pregnancies, tele-ultrasound interpretation, tele-abortion services, and digital patient education platforms have been implemented to various degrees globally [21,22,23]. These interventions reduce patient burden, improve care continuity, and support the early detection of complications [22,24].

Empirical evidence affirms that telemedicine interventions in OB/GYN improve maternal and fetal outcomes, especially among high-risk pregnancies, by facilitating timely monitoring and multidisciplinary consultation [22,24]. For example, studies of telemedicine for medical abortion in Australia report patient satisfaction rates between 78% and 92%, highlighting benefits such as privacy, convenience, and reduced travel [23,25]. During the pandemic, telehealth models ensured the safe and effective management of gestational hypertension and diabetes, achieving outcomes comparable to in-person care [24]. Patient and clinician satisfaction with prenatal telemedicine consultations was generally high, reflecting the growing acceptance of virtual perinatal care [26,27].

The rapid integration of telemedicine in OB/GYN care settings required operational adaptations, including workflow redesign and clinician training [27,28,29]. Variations in physician telemedicine provision are influenced by factors such as specialty, geography, and patient demographics [30]. Data suggest that rural patients more often utilize voice or SMS-based consultations, while urban populations favor video-enabled platforms, emphasizing the need for adaptable technological solutions to meet diverse patient preferences [31]. Telemedicine also plays a growing role in medical education, facilitating clinical exposure and training continuity amid social distancing [32,33].

Looking forward, telemedicine’s sustained adoption depends on addressing persistent infrastructure gaps, regulatory challenges, and digital literacy barriers, especially in resource-constrained settings [34,35,36]. Strategies involving community engagement, policy reform, and the integration of AI and mHealth technologies promise to optimize service delivery and equity [37,38,39]. The continuous evaluation of telemedicine quality, safety, and patient satisfaction remains essential to refining these models and mitigating potential disparities [39,40,41].

This review aims to synthesize current evidence on telemedicine adoption, effectiveness, barriers, and technological innovations in obstetrics and gynecology across diverse global contexts, highlighting both successes and ongoing challenges in this evolving field.

## 2. Materials and Methods

### 2.1. Study Design

This scoping review was conducted following the Preferred Reporting Items for Systematic Reviews and Meta-Analyses extension for Scoping Reviews (PRISMA-ScR) guidelines. The approach enabled a comprehensive mapping of telemedicine applications within obstetrics and gynecology (OB/GYN) care, focusing on adoption, clinical effectiveness, technological integration, and equity across diverse healthcare settings. As a secondary research study, it synthesizes evidence from the existing literature without involving primary data collection or clinical interventions.

### 2.2. Information Sources and Search Strategy

A systematic literature search was performed across multiple scientific databases, including PubMed, Scopus, and Google Scholar. Additionally, targeted manual searches were conducted in high-impact journals such as *The New England Journal of Medicine*, *The Lancet*, *JAMA*, *PLoS ONE*, and *SAGE* digital archives to capture relevant studies potentially not indexed in broader databases. The search covered publications from January 2010 to December 2023.

Search terms combined Medical Subject Headings (MeSH) and free-text keywords, including “telemedicine”, “telehealth”, “obstetrics”, “gynecology”, “digital health”, “rural healthcare”, “artificial intelligence”, and “wearable devices.” Boolean operators (AND, OR) and filters were applied to refine the results to peer-reviewed journal articles. Reference lists of included studies were screened to identify the additional relevant literature. These studies are summarized in Table A1 (see Appendix A). The full search strategy for each database is detailed in Appendix B. The initial search yielded 447 articles. After applying inclusion and exclusion criteria, 142 articles were selected for screening. Following exclusion by title and abstract, 94 articles were chosen for full-text review. Finally, 63 articles were included for data extraction. A PRISMA flowchart (Figure A1) illustrates the study selection process (see Appendix C).

### 2.3. Eligibility Criteria

Studies were eligible if they met the following criteria:Published in English in peer-reviewed journals between 2010 and 2023.Focused explicitly on telemedicine or digital health interventions within obstetrics and/or gynecology.Reported empirical data from rural and/or urban healthcare settings.Included application or evaluation of advanced technologies such as artificial intelligence (AI), mobile health (mHealth), or wearable devices.For COVID-19-era studies (2020–2023), explicit reference to the pandemic context or impact was required.

Exclusion criteria included the following:Publications not in English.Studies focusing exclusively on telemedicine in unrelated specialties.Non-empirical works such as reviews without original data.

### 2.4. Study Selection and Data Extraction

Two independent reviewers screened titles and abstracts for relevance, followed by a full-text assessment against the inclusion criteria. Discrepancies were resolved through discussion or consultation with a third reviewer. Inter-rater reliability was evaluated using Cohen’s kappa coefficient (κ = 0.84), indicating substantial agreement.

A standardized data extraction form was developed and pilot-tested to capture key study characteristics, including author information, publication year, geographic setting, study design, healthcare context, telemedicine modalities, technological components, implementation outcomes, barriers and facilitators, and insights related to COVID-19 adaptations.

### 2.5. Quality Appraisal

Included studies were appraised for methodological quality using validated critical appraisal tools appropriate to their study designs (e.g., CASP checklists for qualitative studies, Joanna Briggs Institute tools for observational studies). Quality assessment results are summarized in Table A2 (see Appendix D).

### 2.6. Data Synthesis and Analysis

Thematic synthesis was conducted using NVivo 14 2023 software to inductively identify recurrent themes across studies. Themes were categorized into overarching domains, including telemedicine benefits, challenges, equity of access, technological readiness, and clinical effectiveness. Comparative analyses explored variations between rural and urban settings and examined the role of AI and wearable devices in enhancing remote OB/GYN care.

A subgroup analysis focused on studies published during the COVID-19 pandemic (March 2020–December 2023), assessing how pandemic-related factors influenced telemedicine adoption, service continuity, and policy innovations.

### 2.7. Ethical Considerations

As this study involved a secondary analysis of the published literature without direct human participant involvement, ethical approval was not required. All sources were cited appropriately, and the review adhered to PRISMA-ScR guidelines to ensure transparency and reproducibility.

### 2.8. Limitations

This review is limited by the exclusion of the non-English literature, potentially underrepresenting non-English-speaking regions. Publication bias is a concern, as unsuccessful or unpublished telemedicine initiatives may be underreported. Additionally, heterogeneity in study designs and outcome measures limited opportunities for quantitative synthesis.

## 3. Results

### 3.1. COVID-19 as a Catalyst for Structural Transformation

The onset of COVID-19 triggered an unprecedented surge in tele-OB/GYN services. Utilization increased by over 500% during the early pandemic period [13], with U.S. OB/GYN visits conducted virtually rising from <1% to 17% [24], and 82% of Indian obstetricians adopting telecare by mid-2020 [15]. In the Middle East, widespread uptake was similarly reported [14], while digital health investment soared by 1818% to USD 788 million in Q1 2020 alone [39]. In Latin America, previously absent platforms saw rapid adoption, transforming telehealth into a structural component of care [16]. By 2022, hybrid care models were adopted by 71% of OB/GYN clinics in high-income countries [42], with routine remote consultations accounting for 9–12% of services [43], signaling a shift from emergency response to sustained delivery mode [12] (Table 1).

### 3.2. Modalities and Technological Innovations

Tele-OB/GYN services span five main modalities. The most widespread—video and phone consultations—accounted for 60–90% of antenatal and postnatal visits during the pandemic’s peak across India, the U.S., and several Middle Eastern countries [1,5,10,11,24,27,30,44,45]. Remote monitoring innovations—such as home BP cuffs and wearable fetal monitors—enabled the early detection of preeclampsia, increasing detection rates by 22% [3,16,21,22]. In Brazil, nearly half of patients with gestational diabetes used mobile glucose tracking [46], while in Canada, virtual monitoring improved adherence in high-risk pregnancies [23].

Artificial intelligence (AI) technologies have enhanced diagnostic capacity. AI-supported triage and imaging tools achieved 89–92% concordance for fetal anomalies and gynecologic cancers [7,8,20,39,47]. Labor admission AI tools shortened triage times by 35% [48]. Remote procedural supervision, such as guided ultrasound in rural India and Ethiopia, reduced unnecessary referrals by 31% [26,29,34]. Store-and-forward platforms supported 20–35% of asynchronous consults in Ghana, Spain, and Canada, with diagnostic concordance above 80% [5,6,13,49] (Figure 1, Figure 2 and Figure 3).

### 3.3. Utilization Patterns and Health System Integration

National policy support and infrastructure significantly influenced uptake. In Saudi Arabia, app-based reminders improved OB/GYN appointment adherence from 68% to 89% [38], while Nigeria reported only 29.3% telehealth availability in tertiary centers [35]. Ethiopia’s integration into smart health platforms reduced no-show rates by 30% [7], and Ghana’s expanded telehealth network improved emergency OB/GYN responses by 40% [34,50].

Adolescent reproductive services also transitioned online. In Kenya and the U.S., 41% of young women preferred virtual consultations for contraception [51,52], while mobile apps in Saudi Arabia and Brazil raised appointment adherence by up to 42% [37,46] (Table 1).

### 3.4. Health Equity and Digital Divide

Barriers to equitable access remain substantial. In Uganda, 57% of pregnant women missed virtual antenatal appointments due to unreliable connectivity [53], and 43% of rural Sub-Saharan African facilities lacked sufficient internet [36,54]. A study in North Carolina found that 15% of OB/GYN patients missed virtual visits due to a lack of devices [27], while urban Senegalese women were 2.7× more likely to access telehealth than their rural counterparts [34].

Language, literacy, and cultural dynamics compound inequity. In the U.S., non-English speakers were 2.2× less likely to complete video visits [55], and in Arab countries, 35% of women avoided teleconsultation due to privacy and modesty concerns [56]. Transgender individuals (47%) and refugee populations in Lebanon reported discomfort or fear in using tele-OB/GYN platforms [18,57]. Racial disparities were evident in the U.S., where Black and Latinx women were 30–36% less likely to complete video consultations [19,28,56] (Table 1; Figure 3).

### 3.5. Clinical Effectiveness and Health Outcomes

Tele-OB/GYN services showed comparable outcomes to in-person care in several domains. Studies in Canada, the U.S., and Brazil reported no significant differences in blood pressure control, gestational weight gain, or neonatal outcomes [3,24,40,43,46]. In Bangladesh and Nigeria, digital maternal mental health tools reduced PHQ-9 scores by an average of 3.1 points [4,25,31]. Behavioral nudges (e.g., SMS reminders) increased postpartum follow-up compliance by 26% and breastfeeding initiation by 15–20% in Saudi Arabia and India [11,14,22,38].

However, exclusive virtual models may risk care quality. A study of OB/GYN oncology patients found that 27% experienced delays in diagnosis or therapy when using remote-only consultations [41,58], and some high-risk pregnancies lacked timely escalation [12] (Table 1; Figure 3).

### 3.6. Provider Experience and Readiness

Provider acceptance is generally favorable, with 80% of OB/GYN clinicians in Saudi Arabia and the U.S. reporting positive experiences [2,5,15,28,29]. Nonetheless, 41–68% cited fatigue, loss of nonverbal cues, and reduced patient connection [5,15]. Diagnostic concordance for OB/GYN conditions in virtual settings was 89.5% at the Mayo Clinic—lower than for dermatology or psychiatry [59]—and 7% of fetal distress cases in India were missed during remote triage [27].

Training remains inadequate in many settings. While 95% of U.S. residents were exposed to telemedicine, only 48% felt prepared for emotionally complex OB/GYN consults [17,32,33]. In Pakistan, fewer than 25% of OB/GYN students received formal training [60], limiting sustainable adoption (Table 1; Figure 3).

### 3.7. Policy, Regulation, and Sustainability

Policy frameworks strongly shaped telemedicine’s sustainability. Only 30% of Latin American countries had comprehensive telehealth legislation by 2022 [61,62], and state licensure laws in the U.S. created barriers to cross-state OB/GYN care [61]. Post-pandemic policy retraction in Canada led 74% of OB/GYNs to scale back virtual offerings [13,39].

Privacy concerns persisted across regions. In Europe and Africa, 28% and 43% of OB/GYN patients, respectively, reported reluctance due to confidentiality fears [9,63]. These legal and ethical limitations continue to hinder full-scale digital integration (Table 1; Figure 3).

**Table 1 healthcare-13-02036-t001:** Thematic summary of telemedicine in obstetrics and gynecology: applications, outcomes, and evidence sources.

Theme	Telemedicine Modality/Key Findings	Clinical or System Outcomes	Geographic Focus	References
Modalities and Innovations	Synchronous (video/audio), asynchronous (store-and-forward), remote monitoring, AI triage, tele-guided ultrasound	+89% diagnostic concordance (AI); 35% faster triage; 22% improved hypertension detection	US, India, Brazil, Ethiopia	[1,14,15,22,26,27,34,46,50]
Utilization and Adoption	>500% increase during COVID-19; sustained 9–12% post-pandemic use; <30% coverage in some LMICs	Improved continuity; increased prenatal visit adherence	US, UK, Nigeria, Saudi Arabia	[5,13,14,24,27,35,38,43]
Equity and Accessibility	Barriers: internet access, language, digital literacy, cultural norms	2.2× lower use in non-English speakers; 57% missed visits due to connectivity; 43% of rural clinics lacked internet	US, Senegal, Lebanon, Uganda, Sub-Saharan Africa	[18,19,27,28,54,55,56,57]
Clinical Effectiveness	Virtual prenatal/postnatal care; SMS/mHealth interventions	Equivalent maternal/neonatal outcomes; −3.1 PHQ-9 depression score; +15–20% breastfeeding rates	US, Brazil, Bangladesh, Nigeria, India	[4,11,14,22,24,31,38,40,46]
Provider Experience	Satisfaction, burnout, training gaps	80% perceived benefit; 41–68% screen fatigue; <50% felt emotionally prepared	US, Saudi Arabia, Pakistan	[2,5,15,28,29,32,33,59]
Policy and Reimbursement	Licensing constraints; legal frameworks; reimbursement; data privacy	Limited cross-border services; 74% reduced telehealth post-funding	US, Latin America, Canada, Europe	[9,13,16,39,54,61,62]
Technology and AI Integration	mHealth apps; AI diagnostics; smart city platforms	+25–42% adherence; >90% cervical cancer AI sensitivity; 34% fewer transfers	Brazil, Saudi Arabia, Ethiopia, Ghana	[7,34,37,38,46,47,48]
COVID-19 as Catalyst	Pandemic-driven expansion; emergency tool implementation	1818% funding increase; sustained hybrid models; 27% oncology treatment delays	Global and US	[24,39,41,42,43,58]
Reproductive Health in Adolescents	Tele-contraceptive and educational services	41% adolescent preference for virtual visits; increased access in underserved areas	Kenya and US	[51,52]

## 4. Discussion

The COVID-19 pandemic acted as a powerful catalyst, accelerating the adoption of telemedicine in obstetrics and gynecology (tele-OB/GYN) worldwide. Our review confirms an explosive growth in tele-OB/GYN services, with utilization increasing by over 500% during the pandemic and sustained adoption rates of approximately 10% in high-income countries [5,13,24,27,43]. This surge demonstrated telemedicine’s capacity to maintain continuity of care amid crisis conditions, particularly for low-risk prenatal and postnatal consultations. However, this rapid expansion also highlighted critical ambiguities regarding the clinical boundaries of virtual care and the risks associated with extending telehealth into diagnosis-intensive and procedural domains.

While tele-OB/GYN facilitated effective remote monitoring and routine care, evidence of diagnostic delays and discordances, albeit sparse, calls for caution. Only a minority of studies reported adverse outcomes such as missed fetal distress or delayed cancer diagnoses [41,58,59], and these were typically framed as secondary findings. The conspicuous absence of focused investigations into telemedicine failures or program discontinuations reveals a publication bias skewed toward success narratives. This gap impairs the field’s ability to critically assess safety thresholds and to develop evidence-based guidelines that integrate risk stratification and hybrid care models optimized for maternal and reproductive health.

Despite promises of democratizing access, tele-OB/GYN has not fully addressed persistent inequities. Structural barriers—limited internet infrastructure, digital illiteracy, language obstacles, and cultural constraints—significantly hindered access, especially for marginalized groups [18,19,27,28,54,55,56]. Our synthesis found that populations in rural Sub-Saharan Africa and South Asia experienced disproportionately high rates of missed telehealth visits due to connectivity issues and sociocultural norms. Similarly, ethnic minorities and non-English speakers in high-income countries faced lower tele-visit completion rates. Vulnerable groups such as transgender individuals and refugees expressed unique psychological and privacy concerns, underscoring the necessity for culturally sensitive, inclusive telehealth platform designs that center equity and user agency.

A major limitation identified is the underrepresentation of fragile and low-resource contexts, which bear the greatest maternal health burdens globally. The paucity of studies from conflict-affected or resource-limited settings impedes the generation of context-specific evidence critical for scaling tele-OB/GYN in these environments. Unique challenges—unstable power supplies, weak digital infrastructure, workforce shortages, and policy gaps—are seldom addressed, rendering most current telehealth frameworks technocentric and ill-suited for such fragile health systems. Intentional inclusion of these settings in future research is essential to develop resilient, context-adaptive telemedicine models.

Tele-OB/GYN has redefined the clinical workflow and provider–patient interactions, but not without unintended consequences. While approximately 80% of providers reported perceived benefits, a substantial proportion experienced screen fatigue, emotional detachment, and heightened diagnostic uncertainty [5,15,17,28,29,32,33,58,59]. Training programs largely emphasize technical competencies, with an insufficient focus on communication skills, ethical dilemmas, and emotional intelligence—skills vital for sensitive maternal care. The psychosocial demands of virtual care, including delivering bad news and managing patient distress, remain underexplored and inadequately addressed in curricula, limiting providers’ holistic preparedness.

The sustainable integration of tele-OB/GYN requires more than technological readiness; legal, regulatory, and reimbursement frameworks remain fragmented. Licensing restrictions constrain cross-border consultations, and inconsistent reimbursement policies lead to reduced telehealth usage post-pandemic emergency funding [13,16,39,61,62]. Concerns about data privacy and security persist, particularly among vulnerable populations, and the lack of robust governance in many low- and middle-income countries hinders the transition from pilot projects to scale [9,54,63]. Addressing these policy gaps with equity-focused, rights-based approaches will be crucial for ethical mainstreaming and long-term sustainability.

The evidence base is characterized predominantly by descriptive studies and program evaluations with variable methodological rigor. Longitudinal designs, randomized controlled trials, and comparative effectiveness research are rare. Moreover, there is a lack of synthesis frameworks distinguishing intervention types, contextual factors, and patient subgroups. For tele-OB/GYN to mature as a research field, future studies must adopt rigorous, thematically structured methodologies capable of elucidating not only what works, but why, how, and for whom. Transparent reporting of both successes and failures will enhance learning and inform adaptive implementation strategies.

Telemedicine in obstetrics and gynecology has demonstrated remarkable potential to extend care, improve outcomes, and enhance patient convenience, particularly during public health emergencies. Yet, the clinical, ethical, and equity challenges illuminated by our review underscore that tele-OB/GYN is not a panacea. Achieving safe, equitable, and sustainable virtual maternal healthcare demands robust evidence, inclusive design, comprehensive provider training, and cohesive policy frameworks. Only through a balanced appraisal of both achievements and shortcomings can telemedicine fulfill its promise to transform maternal health globally.

## 5. Conclusions

Telemedicine in obstetrics and gynecology has rapidly evolved from a crisis-driven innovation to an integral component of contemporary maternal healthcare. This review highlights the diverse modalities underpinning tele-OB/GYN services and their demonstrated potential to enhance access, improve clinical outcomes, and support patient-centered care. However, significant challenges remain, including unresolved clinical boundaries for remote care, critical gaps in addressing implementation failures, and persistent inequities driven by intersecting social determinants.

To realize the full promise of tele-OB/GYN, future research must adopt rigorous, evidence-based methodologies that systematically evaluate both successes and failures across varied settings, with particular attention to low-resource and marginalized populations. The integration of advanced AI-driven tools holds promise but requires careful validation to ensure equity, safety, and acceptability. Furthermore, sustainable expansion depends on cohesive policy frameworks addressing legal, reimbursement, and data privacy concerns.

Ultimately, tele-OB/GYN offers a transformative opportunity to reimagine reproductive healthcare delivery. Achieving this potential demands a holistic, equity-centered approach that balances technological innovation with ethical imperatives and clinical prudence.

## Figures and Tables

**Figure 1 healthcare-13-02036-f001:**
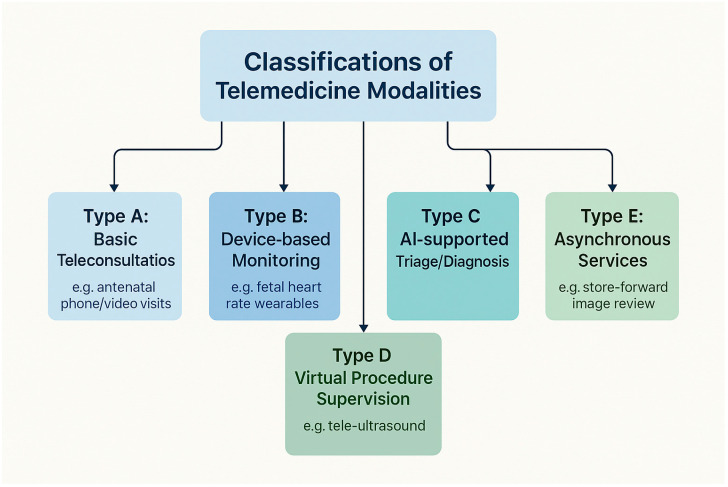
Classification framework of telemedicine modalities in obstetrics and gynecology.

**Figure 2 healthcare-13-02036-f002:**
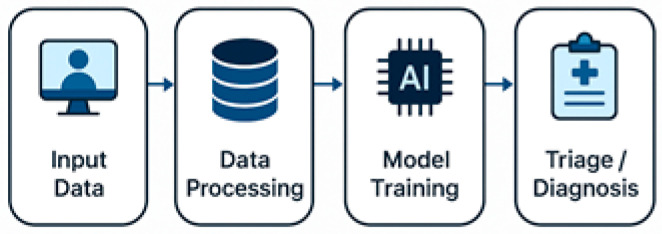
AI-powered triage and diagnosis pipeline in obstetrics and gynecology.

**Figure 3 healthcare-13-02036-f003:**
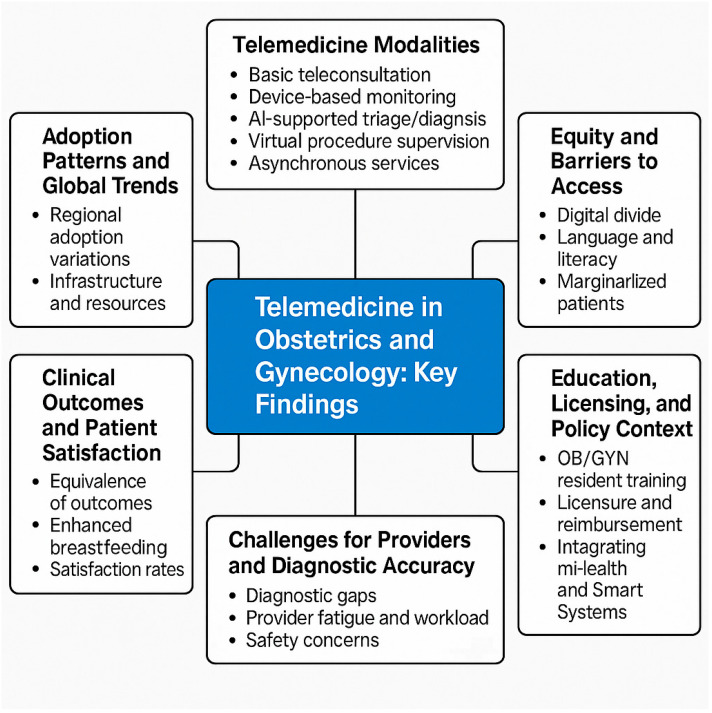
Conceptual framework of telemedicine in obstetrics and gynecology: key themes and findings.

## Data Availability

All data was included in the article.

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
