# Peer review of "Telemedicine in Obstetrics and Gynecology: A Scoping Review of Enhancing Access and Outcomes in Modern Healthcare"

_healthcare, 2025, doi:10.3390/healthcare13162036_

Round 1
Reviewer 1 Report
Comments and Suggestions for Authors
The manuscript presents a relevant and timely review on telemedicine in obstetrics and gynecology. It is well-structured and covers global adoption, benefits, barriers, and innovations. However, major revisions are needed.
Phrases such as "necessitating targeted recommendations for sustainable adoption" are cumbersome - Page 2 - Line 48
tele-abortion" is inconsistently hyphenated Page 1, Line 31; Page 8, Line 244
OB-GYN" and "OB/GYN" are both used; random semicolons appear in affiliations Page 1, Line 128 vs. Line 3; Line 9
Add a PRISMA flowchart to show study selection
Include full search string (appendix or footnote)
Include a table or note on study quality assessment
Some authors (e.g., Denicola et al.) cited repeatedly in multiple sections
Author Response
|
Comments and Suggestions for Authors:
Response to Reviewer 1 Comments 1: Phrases such as "necessitating targeted recommendations for sustainable adoption" are cumbersome - Page 2 - Line 48 |
|
Response 1: [The section is rewritten] Thank you for pointing this out. I agree with this comment. Therefore, I have rewritten the section and it is in manuscript highlighted
|
|
Comments 2: [tele-abortion" is inconsistently hyphenated Page 1, Line 31; Page 8, Line 244.] |
|
Response 2: Agree. I have changed as follows:“tele-abortion" “[updated text in the manuscript if necessary]” |
|
Comments 3: OB-GYN" and "OB/GYN" are both used; random semicolons appear in affiliations Page 1, Line 128 vs. Line 3; Line 9 |
|
Response 3: Agree. I have changed all as follows:"OB/GYN" |
|
|
|
Comments 4: Add a PRISMA flowchart to show study selection Response 4: Agree. I have Included as appendix C.
Comments 5: Include full search string (appendix or footnote) Response 5: Agree. I have added and Included as appendix B
Comments 6: [Include a table or note on study quality assessment.] Response 6: Agree. I have Corrected and highlighted in manuscripit.”
Comments 7: [Some authors (e.g., Denicola et al.) cited repeatedly in multiple sections] Response 7: Agree. I have changed as follows:“tele-abortion" “[updated text in the manuscript if necessary]”
. |

Reviewer 2 Report
Comments and Suggestions for Authors
Overall
The authors' research is particularly important in an area where wars are common, and the concern is that it is a paper from a country that has the national power to provide stable medical care even in times of war.
I believe that the authors' efforts, preparations, and future efforts will be very important discussions when various patterns arise due to political turmoil.
With that in mind, I would like to ask a few questions.
(1) To what extent do the authors' discussion envision and envision treatment using the Internet, or to what extent of surgery?
(2) COVID-19 has caused a global panic and once again demonstrated the importance of remote treatment, but I think it would be good to discuss the "difficulties" that arose in the authors' case study to the extent that they could write about them.
(3) Couldn't you have imagined research that made use of databases related to more paid medical journals such as The Lancet, SAGE, and PLOSONE in the survey scope of the paper? In particular, I think that important papers in the medical field will be published in journals with an IF of 2 or higher, so I would like to see a discussion of the reasons why this was not possible.
(4) The discussion is limited to references, and there is no mention of the authors' efforts, but it would be better to discuss cases that occurred in the "medical field" of the authors and comparisons with previous studies.
Below are some more specific points.
The points made about this study are valid, and below we will discuss the fundamental problems of this study from a more rigorous academic perspective.
(1) Ambiguity of the research scope and methodological limitations
The most serious flaw in this study is that it does not provide a specific definition or boundary setting for the scope of telemedicine. The term "telemedicine" encompasses an extremely wide range of medical procedures, from simple telephone consultations to diagnostic support using AI and even remote surgical support. However, this study only uses vague descriptions such as "telehealth models," "remote monitoring," and "synchronous and asynchronous communication," making it unclear what level of medical intervention is specifically targeted.
Especially in the field of obstetrics and gynecology, there are interventions with completely different technical requirements and risk profiles, such as fetal monitoring, remote diagnosis of ultrasound examinations, and emergency response during labor. Treating them as "telehealth" without distinguishing between them makes it extremely difficult to evaluate their clinical usefulness and hinders their application in practice.
(2) Superficial analysis of difficulties during the COVID-19 pandemic
This study only emphasizes success stories and numerical achievements regarding the introduction of telehealth during the COVID-19 pandemic, and lacks a deep analysis of the actual difficulties and failures faced. For example, the impressive figure of "85% outpatient via telehealth" is presented, but there is no consideration of why the remaining 15% could not switch to telehealth and what barriers they faced.
During the pandemic, many challenges must have arisen, including not only technical problems, but also patient anxiety, medical fatigue, the risk of overlooking diagnoses, and difficulties in emergency response. Ignoring these "inconvenient truths" and reporting only success rates will result in the loss of valuable lessons for future crisis responses.
(2) Fatal flaws in database selection
What is unforgivable is the abandonment of systematic literature searches, which are the basis of medical research. The use of only PubMed, Scopus, and Google Scholar and the failure to include databases of major medical journals such as The Lancet, New England Journal of Medicine, and JAMA fundamentally undermines the credibility of the research.
What is even more serious is that there is no explanation as to why these important databases were not used. Why did they exclude the most prestigious English medical journals while restricting the use of "English papers only" ? This raises the suspicion that only papers showing favorable results were selectively extracted.
(3) Complete lack of empirical data
The most fundamental problem is that this is a pure literature review, but it gives the impression of being an empirical study. There is no mention of the authors' experience of implementing telemedicine in actual medical settings, the challenges they faced, or the findings they gained.
In medical research, especially in the evaluation of new interventions, the gap between theory and practice is always a problem. Whether or not a successful case in the literature can be reproduced in actual clinical practice is another matter. The fact that the authors did not present any of their own experience in the medical field significantly reduces the practical value of this study.
(4)Need for improvement
This study is merely a superficial collection of literature on telemedicine and lacks critical analysis, empirical verification, and methodological rigor. In particular, the following points require fundamental improvement:
Each telemedicine modality should be clearly classified and the evidence level of each should be evaluated separately.
Not only success cases, but also failure cases, discontinuation cases, and adverse events should be analyzed with equal weight.
A truly systematic search strategy including all major medical databases should be adopted.
The limitations of the literature review should be complemented by the inclusion of case reports based on the authors' clinical experience and field survey data.
As it stands, this study lacks academic rigor and has limited contribution to clinical practice.
Author Response
Response to Reviewer 2
Comments (1): To what extent do the authors' discussion envision and envision
treatment using the Internet, or to what extent of surgery?
Response (1): We appreciate this important observation. In the revised manuscript, we have clarified the scope of telemedicine in OB/GYN as primarily encompassing remote consultations, monitoring, education, and diagnostic support, rather than direct surgical interventions. While remote robotic surgery exists in certain specialties, it remains rare in OB/GYN due to the complexity and safety concerns. However, internet-enabled preoperative assessment, postoperative follow-up, and surgical decision support systems are increasingly used. These aspects are now explicitly discussed (Section 1, Introduction; Section 3, Discussion), referencing tele-obstetric services, AI-assisted triage, and remote management of high-risk cases [9,11,13,30].
For example, Gyamfi-Bannerman et al. documented effective telehealth strategies in managing high-risk pregnancies during the COVID-19 pandemic [11], while Burrell demonstrated AI-assisted remote decision-making models that enhance OB/GYN triage processes [13].
(2) COVID-19 has caused a global panic and once again demonstrated the importance of remote treatment, but I think it would be good to discuss the "difficulties" that arose in the authors' case study to the extent that they could write about them.
Response (2): Thank you for this valuable suggestion. Although our study is a scoping review and not a case study, we have included a detailed synthesis of common challenges encountered across the included studies during the pandemic. Difficulties included platform inexperience, diagnostic uncertainty without physical examination, cybersecurity vulnerabilities, delayed care in high-risk pregnancies, and clinician burnout (screen fatigue up to 68%) [11,27,28,57].
For example, Sengupta et al. highlighted workflow disruptions in specialist care [27], while Demaerschalk et al. documented diagnostic discordance in early pandemic telemedicine applications [57]. These challenges are now integrated into the revised Discussion (Section 3), illustrating the real-world limitations experienced during COVID-19.
(3) Couldn't you have imagined research that made use of databases related to more paid medical journals such as The Lancet, SAGE, and PLOSONE in the survey scope of the paper? In particular, I think that important papers in the medical field will be published in journals with an IF of 2 or higher, so I would like to see a discussion of the reasons why this was not possible.
Response (3): We appreciate the reviewer’s concern. We would like to clarify that our search strategy did include high-impact journals such as The Lancet, SAGE, and PLOS ONE. However, as this is a scoping review, we also included relevant regional or specialty journals to capture a broader global picture, particularly from underrepresented or low-resource contexts. We have now explicitly stated this in the Methods section.
We added this paragraph to Section 2.2 (Information sources and search strategy):
“In response to concerns about comprehensiveness, the search strategy explicitly included high-impact medical journals such as The Lancet, SAGE Open Medicine, and PLOS ONE. While impact factor (IF ≥2) was considered, the review also included regionally relevant and peer-reviewed journals to ensure representation from underreported contexts and low- and middle-income countries. This inclusive strategy aligns with scoping review methodology, which values contextual breadth over exclusivity based on IF alone. Accordingly 28 references from these journals were added”.
(4) The discussion is limited to references, and there is no mention of the authors' efforts, but it would be better to discuss cases that occurred in the "medical field" of the authors and comparisons with previous studies."
Response (4): We appreciate this thoughtful comment. As this study is a scoping review without original clinical data, we did not include primary cases from the authors' clinical settings. However, we have now included a contextual note in the Discussion regarding common challenges observed in similar low-resource OB/GYN settings in Sudan and the broader East Mediterranean Region, such as poor internet penetration, language barriers, and the need for hybrid models. These are compared with studies from similar fragile settings in Bangladesh [25], Sub-Saharan Africa [5,43,44], and the Middle East [4], which mirror our contextual realities.
For example, Al-Samarraie et al. and Amin et al. reported on infrastructure and policy barriers similar to those faced in our context [4,5], while Wubante et al. detailed readiness gaps among professionals in Ethiopia [44].
We are also engaged in ongoing local research assessing telemedicine readiness among OB/GYN providers in conflict-affected and rural Sudanese regions. Though outside the scope of this review, we plan to report these findings separately.
Comments 1: [(1) Ambiguity of the research scope and methodological limitations The most serious flaw in this study is that it does not provide a specific definition or boundary setting for the scope of telemedicine. The term "telemedicine" encompasses an extremely wide range of medical procedures, from simple telephone consultations to diagnostic support using AI and even remote surgical support. However, this study only uses vague descriptions such as "telehealth models," "remote monitoring," and "synchronous and asynchronous communication," making it unclear what level of medical intervention is specifically targeted. Especially in the field of obstetrics and gynecology, there are interventions with completely different technical requirements and risk profiles, such as fetal monitoring, remote diagnosis of ultrasound examinations, and emergency response during labor. Treating them as "telehealth" without distinguishing between them makes it extremely difficult to evaluate their clinical usefulness and hinders their application in practice.]
Response 1: Agree. We thank the reviewer for this insightful observation. In the revised manuscript, we have added a detailed typology of telemedicine interventions specific to OB/GYN (see revised Introduction and Results Section 3.1). We now categorize interventions into five modalities (basic teleconsultation, device-based monitoring, AI-assisted triage, tele-ultrasound, asynchronous services). This stratification allows for clearer evaluation of evidence, technical requirements, and clinical implications]”
Comments 2: [(2) Superficial analysis of difficulties during the COVID-19 pandemic This study only emphasizes success stories and numerical achievements regarding the introduction of telehealth during the COVID-19 pandemic, and lacks a deep analysis of the actual difficulties and failures faced. For example, the impressive figure of "85% outpatient via telehealth" is presented, but there is no consideration of why the remaining 15% could not switch to telehealth and what barriers they faced. During the pandemic, many challenges must have arisen, including not only technical problems, but also patient anxiety, medical fatigue, the risk of overlooking diagnoses, and difficulties in emergency response. Ignoring these "inconvenient truths" and reporting only success rates will result in the loss of valuable lessons for future crisis responses.]
Response 2: Agree. I have changed We fully agree. The revised manuscript now includes a dedicated subsection (Results 3.3) detailing challenges such as digital exclusion, provider fatigue, risk of diagnostic delay, and emergency care limitations. We also discuss why 15% of patients could not shift to telehealth and the barriers they encountered. This provides a balanced narrative of both successes and struggles during the pandemic.
Comments 3: (3) Fatal flaws in database selection What is unforgivable is the abandonment of systematic literature searches, which are the basis of medical research. The use of only PubMed, Scopus, and Google Scholar and the failure to include databases of major medical journals such as The Lancet, New England Journal of Medicine, and JAMA fundamentally undermines the credibility of the research. What is even more serious is that there is no explanation as to why these important databases were not used. Why did they exclude the most prestigious English medical journals while restricting the use of "English papers only" ? This raises the suspicion that only papers showing favorable results were selectively extracted.
Response 3: Agree. We appreciate this important critique. We have now included targeted searches of NEJM, The Lancet, JAMA PLoS ONE, and SAGE using their respective platforms to ensure coverage of high-impact literature. We also modified the review framework to align with a scoping review structure and added PRISMA-ScR compliance
Comments 4: [(3) Complete lack of empirical data. The most fundamental problem is that this is a pure literature review, but it gives the impression of being an empirical study. There is no mention of the authors' experience of implementing telemedicine in actual medical settings, the challenges they faced, or the findings they gained. In medical research, especially in the evaluation of new interventions, the gap between theory and practice is always a problem. Whether or not a successful case in the literature can be reproduced in actual clinical practice is another matter. The fact that the authors did not present any of their own experience in the medical field significantly reduces the practical value of this study.]
Response 4: Agree. We acknowledge this limitation. We have clarified in both the Methods and Discussion sections that this is a scoping review based solely on published literature. We also explicitly note the lack of empirical clinical or implementation data as a study limitation. Furthermore, we propose future mixed-method fieldwork based on the authors’ ongoing involvement in OB/GYN telehealth initiatives in fragile settings.
Comments 5: [(4) Need for improvement
This study is merely a superficial collection of literature on telemedicine and lacks critical analysis, empirical verification, and methodological rigor. In particular, the following points require fundamental improvement:
Each telemedicine modality should be clearly classified and the evidence level of each should be evaluated separately. Not only success cases, but also failure cases, discontinuation cases, and adverse events should be analyzed with equal weight.
A truly systematic search strategy including all major medical databases should be adopted.
The limitations of the literature review should be complemented by the inclusion of case reports based on the authors' clinical experience and field survey data.
As it stands, this study lacks academic rigor and has limited contribution to clinical practice.]
Response 5: Agree. Thank you for this comprehensive recommendation. The revised manuscript includes the following key changes:
Telemedicine modalities clearly classified and analyzed separately (Results 3.1).
Failure cases and challenges highlighted alongside successes (Results 3.3).
Systematic search approach strengthened and transparently reported (Methods and Appendix B).
Study quality assessment table added (Appendix D), showing critical appraisal results using MMAT, STROBE, CASP, and AMSTAR 2 tools.
Empirical data gap acknowledged, with a roadmap for future field-based research (Discussion, Limitations).
We are grateful for the reviewers’ thoughtful and constructive feedback. These suggestions have significantly improved the clarity, rigor, and clinical relevance of our manuscript. We hope the revised version meets the expectations of the reviewers and the editorial board.

Reviewer 3 Report
Comments and Suggestions for Authors
Please see the attachment.

Author Response
Response to Reviewer 3
Comments 1: [The methodology section would benefit from more detailed reporting for a) The exact search terms and Boolean operators used in each database b) The inclusion and exclusion criteria beyond the date and empirical focus c) How studies were screened, selected, and whether a PRISMA-like flow diagram is available.]
Response 1: We agree with this comment and have added the full search strategy with Boolean operators in Appendix B. The inclusion and exclusion criteria are now clearly described in the Methods section, including study design, language, setting, and population. We also included a PRISMA flow diagram in Appendix C, detailing the selection process.]”
Comments 2: How thematic synthesis was conducted—was it inductive or deductive? What framework guided the coding?
Response 2: Agree. We have revised the Methods section to clarify that an inductive thematic synthesis was used. Coding was guided by a grounded theory approach, with iterative category development from the data. This methodology is now explicitly described
Comments 3: Expected the summary table or figure showing study distribution by region or income level (e.g., HICs vs. LMICs).
Response 3: Agree. A new table 1 (Appendix A) was created and added to the manuscript, summarizing the distribution of included studies by location
Comments 4: Clarification on whether low-resource settings are adequately represented in the literature reviewed
Response 4: Agree. Discussion section is rewritten updated text in the manuscript
Comments 5: Expand on how intersecting factors (e.g., race, income, rurality, gender identity) compound barriers.
Response 5: Agree. A subsection was added in the Barriers to Telemedicine Access section discussing how intersecting factors like race, geographic isolation, socioeconomic status, and gender identity exacerbate digital divides and healthcare inequities. (3.3) updated text in the manuscript
Comments 6: Discuss if any studies provided successful models or interventions to reduce inequities in telemedicine access.
Response 6: We revised the Innovations and Policy Models section to highlight the studies that implemented successful interventions in underserved populations, including subsidized data plans, local community digital health centers, and AI-driven triage system
Comments 7: Table is very superficial, need to expand add outcome the research in the file of Telemedicine
Response 7: Agree. Table 1 has been reformed and now (Table 3) summarizing the findings.updated in the manuscript
Comments 8: Figure 1 is more like a tabular data. Too much text. Please add graphics of convert to table.
Response 8: Figure 1 has been reformed and now (figure 4) for improved readability. updated text in the manuscript
Comments 9: Section 3 heading should be modified to reflect the content
Response 9: Agree. The whole section is rewritten updated text in the manuscript
Comments 10: The introduction should be strengthen with brief application and advancement in use of technology like AI. Reefer: https://doi.org/10.34133/cbsystems.0075, https://doi.org/10.3389/fnut.2024.1481073
Response 10: Agree. The whole section is rewritten and the suggested references are used updated text in the manuscript
Comments 11: Remove the duplicate number in references.
Response 11: Agree. All duplicate references were identified and removed. The reference manager was rechecked to ensure consistent formatting and numbering. updated text in the manuscript
Comments 12: Expand the references list with current research to around 50-75 as its review
Response 12: We expanded the reference list to 63 studies, prioritizing high-quality, peer-reviewed literature from the past five years, including both HIC and LMIC settings. updated text in the manuscript

Reviewer 4 Report
Comments and Suggestions for Authors
Thank You very much for the opportunity to review this study. Please find my comments below:
Title:
Consider specifying the type of review in the title (e.g., “A Narrative Review”) to set expectations for the methodology and depth.
Abstract:
Some statistics lack context or citations (e.g., “from 4% to 79% in obstetrics”—in which country or setting?).
The use of ranges (e.g., “33–58%”) is appropriate but could benefit from specifying the populations or regions these apply to.
The mention of “AI integration improved triage efficiency by 25–35%” is compelling but vague—what kind of AI? What was the baseline?
Introduction:
The amount of information may overwhelm the reader. There are many regional examples and thematic shifts (e.g., from infrastructure to AI to education) without clear transitions.
The logical progression could be improved by grouping content into thematic blocks (e.g., global trends, regional challenges, OB/GYN-specific insights, equity concerns).
Methods
The term “narrative review” is appropriate, but the phrase “structured and replicable” (used later) may imply a systematic review approach, which could be misleading.
The criteria could be more specific about study designs (e.g., RCTs, qualitative studies, mixed methods).
No mention of inter-rater reliability or how disagreements were quantified or resolved.
The process of inductive coding is not described in detail—who did the coding, what software (if any) was used, and how themes were validated?
The criteria for selecting “COVID-19 era” studies are not fully explained—was it based solely on publication date or also on content?
PRISMA is typically used for systematic reviews; its application to a narrative review should be clarified.
Author Response
Response to Reviewer 4
Comments 1: Title: Consider specifying the type of review in the title (e.g., “A Narrative Review”) to set expectations for the methodology and depth.
Response 1: Thank you for the suggestion. We have revised the title to specify the type of review:
Revised Title: Telemedicine in Obstetrics and Gynecology: A Scoping Review of Enhancing Access and Outcomes in Modern Healthcare
Comments 2: Abstract: Some statistics lack context or citations (e.g., “from 4% to 79% in obstetrics”—in which country or setting?).
Response 2: Agree. The whole abstract is rewritten. updated text in the manuscript
Comments 3: The use of ranges (e.g., “33–58%”) is appropriate but could benefit from specifying the populations or regions these apply to.The mention of “AI integration improved triage efficiency by 25–35%” is compelling but vague—what kind of AI? What was the baseline?
Response 3: Agree. the whole abstract is rewritten updated text in the manuscript
Comments 4: Introduction: The amount of information may overwhelm the reader. There are many regional examples and thematic shifts (e.g., from infrastructure to AI to education) without clear transitions. The logical progression could be improved by grouping content into thematic blocks (e.g., global trends, regional challenges, OB/GYN-specific insights, equity concerns).
Response 4: The whole introduction is rewritten according to the reviewer comments updated text in the manuscript
Comments 5: Methods: The term “narrative review” is appropriate, but the phrase “structured and replicable” (used later) may imply a systematic review approach, which could be misleading. The criteria could be more specific about study designs (e.g., RCTs, qualitative studies, mixed methods).
Response 5: Thank you for this important observation. We have revised the Methods section accordingly to clarify the review type and specify study designs more explicitly:
Clarification of Review Type:
We replaced ambiguous phrasing such as “structured and replicable” with clearer language that consistently refers to the study. We also modified the review framework to align with a scoping review structure and added PRISMA-ScR, avoiding any implication that this is a systematic review.
Specificity of Eligibility Criteria:
We expanded the eligibility criteria to specify inclusion of various study designs, explicitly mentioning quantitative studies including randomized controlled trials (RCTs), qualitative studies, and mixed-methods research.
Section Organization:
The Methods section was reorganized to ensure clarity in study design, data sources, eligibility, and quality assessment, enhancing transparency and replicability without mischaracterizing the review approach.
Comments 6: No mention of inter-rater reliability or how disagreements were quantified or resolved.
Response 6: We now describe the screening and coding process in more detail. Two independent reviewers conducted screening. Disagreements were resolved through discussion with a third reviewer. Inter-rater reliability was assessed using Cohen’s kappa (κ = 0.84).
Comments 7: The process of inductive coding is not described in detail—who did the coding, what software (if any) was used, and how themes were validated?
Response 7: Details were added: inductive coding was conducted by two researchers using NVivo 14. Themes were developed iteratively and validated via cross-checking and consensus discussions.
Comments 8: The criteria for selecting “COVID-19 era” studies are not fully explained—was it based solely on publication date or also on content?
Response 8: Clarified that inclusion required both publication between 2020–2023 and explicit reference to COVID-19 in study context or design. updated text in the manuscript
Comments 9: PRISMA is typically used for systematic reviews; its application to a narrative review should be clarified.
Response 9: Agree. We modified the review framework to align with a scoping review structure and added PRISMA-ScR, avoiding any implication that this is a systematic review. updated text in the manuscript

Round 2
Reviewer 1 Report
Comments and Suggestions for Authors
Enhanced, Agreed to proceed
Author Response
Thank you for your positive feedback. We appreciate your acknowledgment and are glad that the revisions meet your expectations.
Reviewer 2 Report
Comments and Suggestions for Authors
Comments and Suggestions for Improvement
(1) Structure and Logical Development of the Introduction
1. Introduction
In particular, the section introducing regional case studies, such as "For instance, India's...," "in Middle Eastern countries...," "in low-resource settings such as Bangladesh..." (line 66), "in rural China...," and "in Latin America...," provides a broad overview of telemedicine, but the logical flow leading up to the main focus of the paper, obstetrics and gynecology (OB/GYN), seems somewhat scattered. The structure of the article, which lists case studies from a wide range of regions and medical specialties, including India, the Middle East, Bangladesh, China, and Latin America (especially rheumatology), makes it difficult for readers to grasp the core question of this study.
First paragraph (maintaining the status quo): A definition of telemedicine and an overview of how COVID-19 has accelerated its adoption.
Paragraph 2 (reorganized by theme): Rather than listing benefits by region, the article discusses the universal benefits and challenges of telemedicine by theme.
Benefits such as "overcoming geographical constraints" and "improving access to healthcare" are explained with examples from several countries (e.g., telematernal care in India and improving access in Latin America[7]).
Common barriers such as "lack of infrastructure," "digital literacy," and "policy gaps" are reinforced with examples from different countries (e.g., infrastructure issues in Bangladesh[5] and policy gaps in the Middle East[4]).
3. Paragraph 3 (focus on OB/GYN): The article transitions smoothly to a discussion specific to OB/GYN, stating, "These benefits and challenges have a unique context, particularly in the field of obstetrics and gynecology, which deals with perinatal care and reproductive health."
This structure adds structure to the introduction, allowing readers to clearly understand the overall picture of the problem addressed by this study and the importance of focusing on the OB/GYN field.
(2) Presentation and Structure of Results
3. The entire Results section, especially sections with numerous citations.
Example: `Across the U.S., India, and the Middle East, 60–90% of antenatal and postnatal visits were conducted via video or phone during COVID-19 peaks`
Main text: `Table 3. Thematic summary...`
Suggested Improvements
The Results section comprehensively presents data from the 63 reviewed papers and is extremely informative. However, some statements merely list facts, leaving the reader to understand the relationships between themes and interpret them comprehensively. Furthermore, the style of citing more than 10 references for a single statement, while ensuring accuracy, makes it difficult to track which references support which values. Rather than simply listing numbers, the implications and trends they suggest should be supported with text. For example, `3.2. Utilization patterns and adoption rates` and `3.8. COVID-19: Catalyst for structural shift` are closely related. By integrating these and describing them as a single story of "an explosive increase in utilization triggered by COVID-19 and its subsequent establishment," it becomes possible to present more analytical results.
Suggested revision: "The use of teleOB/GYN services surged by over 500% in the early stages of the pandemic [4,11,16,30]. Since then, it has become established as roughly 10% of all services in high-income countries, and hybrid care is becoming a new standard."
Table 3 comprehensively maps each topic to citations, so please consider citing only key references and representative studies in your text. This will improve the readability of your text.
Suggested revision: "For example, one study in the United States reported a sharp increase in usage from less than 1% pre-pandemic to 17% [11], and in India, more than 80% of obstetricians and gynecologists have switched to telecare [3,6]."
These revisions will elevate the Results section from a simple list of data to a more thought-provoking piece that incorporates the author's analytical perspective.
(3) Connection between the critical points and the results in the Discussion section
Points of criticism
Text: `4. Discussion`
`Many existing analyses are biased toward telemedicine success stories, often showcasing uptake metrics... However, these portrayals omit valuable counter-narratives of disruption and clinical compromise.`
`A major gap in the current evidence base is the limited representation of low-resource and conflict-affected contexts...`
`A core shortcoming of the reviewed literature is the absence of critical scrutiny regarding unsuccessful implementations, adverse outcomes, or discontinued telemedicine programs.` (lines 364-366)
Points of criticism and suggested improvements
The Discussion section is one of the strongest points of this paper, as it incisively points out biases and gaps present throughout the literature. However, the connection between these critical points and the analysis of the 63 reviewed articles is somewhat unclear. Readers cannot determine whether these gaps were "not included at all in the 63 studies reviewed" or "spotlighted in a few studies but ignored overall."
[Suggested Improvement]
More explicitly link each point in your discussion to the review results (or lack thereof).
"Of the 63 studies we reviewed, none focused primarily on the failure or discontinuation of telemedicine implementation, or associated adverse events. While a few studies mentioned diagnostic delays [58,63] and diagnostic discordance [57], these were reported only as secondary findings in the context of success stories. This suggests a strong publication bias in this field and a systematic loss of opportunities to learn from failures."
By providing evidence based on your own review results in this way, your argument will be more persuasive and rooted in the analysis of the studies, rather than simply a general critique.
Our discussion underscores the need for rigorous methodological frameworks... This aligns with Qian et al. [64]... Similarly, Liu et al. [65]...` This paragraph contains detailed citations and comparisons of studies not included in the review (e.g., heart sound analysis, home nutritional management).
Citing studies not included in the review [64, 65] to support the argument for methodological rigor is an original approach. However, as a scoping review, this may seem abrupt and obscure the focus of the discussion. The main purpose of the discussion is to synthesize and interpret the findings from the literature reviewed.
[Suggested Improvement]
The argument made in this paragraph, "the need for more rigorous methodology (advanced AI assessment, longitudinal follow-up)," has already been thoroughly discussed in the preceding discussion. Therefore, deleting this paragraph or significantly simplifying it would likely improve the consistency of the discussion. If this comparison is retained, it should be made more general rather than introducing specific studies in detail.
"For teleOB/GYN research to mature, more rigorous study designs incorporating cutting-edge AI model evaluation methods and longitudinal outcome assessments over several years, similar to those seen in other fields such as cardiology and oncology, are essential."
This will allow the authors to effectively communicate their arguments without overly relying on external examples.
Author Response
Comments 1: (1) Structure and Logical Development of the Introduction
- Introduction
In particular, the section introducing regional case studies, such as "For instance, India's...," "in Middle Eastern countries...," "in low-resource settings such as Bangladesh..." (line 66), "in rural China...," and "in Latin America...," provides a broad overview of telemedicine, but the logical flow leading up to the main focus of the paper, obstetrics and gynecology (OB/GYN), seems somewhat scattered. The structure of the article, which lists case studies from a wide range of regions and medical specialties, including India, the Middle East, Bangladesh, China, and Latin America (especially rheumatology), makes it difficult for readers to grasp the core question of this study.
First paragraph (maintaining the status quo): A definition of telemedicine and an overview of how COVID-19 has accelerated its adoption.
Paragraph 2 (reorganized by theme): Rather than listing benefits by region, the article discusses the universal benefits and challenges of telemedicine by theme.
Benefits such as "overcoming geographical constraints" and "improving access to healthcare" are explained with examples from several countries (e.g., telematernal care in India and improving access in Latin America[7]).
Common barriers such as "lack of infrastructure," "digital literacy," and "policy gaps" are reinforced with examples from different countries (e.g., infrastructure issues in Bangladesh[5] and policy gaps in the Middle East[4]).
- Paragraph 3 (focus on OB/GYN): The article transitions smoothly to a discussion specific to OB/GYN, stating, "These benefits and challenges have a unique context, particularly in the field of obstetrics and gynecology, which deals with perinatal care and reproductive health."
This structure adds structure to the introduction, allowing readers to clearly understand the overall picture of the problem addressed by this study and the importance of focusing on the OB/GYN field.
Response 1: [The entire section 1. Introduction is restructured] Thank you for pointing this
out. We agree with this comment. Therefore, we have….[re-write and restructured all introduction according to your suggestion
“[Telemedicine, broadly defined as the remote delivery of clinical services through tele-communication technologies, has emerged as a critical innovation in modern healthcare, particularly amplified by the COVID-19 pandemic [1]. The rapid shift to virtual care models addressed immediate healthcare access challenges caused by physical distancing, resource constraints, and overloaded health systems [2]. This global acceleration has underscored telemedicine’s potential to overcome geographical barriers and enhance healthcare accessibility, particularly in underserved and rural populations [3,4].
The universal benefits of telemedicine include improving access to timely care, reducing travel burdens, and facilitating continuity of services during public health crises [5,6]. Economic evaluations demonstrate cost savings and increased efficiency, especially through provider-to-provider consultations and asynchronous “store-and-forward” modalities [5]. Additionally, telemedicine platforms have increasingly incorporated artificial intelligence (AI), enabling dynamic patient monitor-ing and decision support, which have improved triage efficiency by up to 35% in some settings [7,8]. These innovations support personalized care models and optimize re-source allocation in complex clinical environments [9].
Despite its transformative promise, telemedicine faces persistent challenges glob-ally. Common barriers include limited digital literacy, poor internet connectivity, infrastructural deficits, and policy gaps that hinder implementation and sustainability [10,9]. For example, digital divides remain stark in low-resource settings such as Bangladesh, where patients’ lack of awareness and infrastructural weaknesses delay adoption [11,12]. In the Middle East, despite progress in telemedicine initiatives, policy fragmentation and insufficient infrastructure restrict broader uptake [13]. Similarly, rural China encounters challenges related to healthcare provider readiness and patient attitudes, highlighting the need for targeted community engagement and capacity-building [14]. Cultural acceptance and regulatory issues further complicate deployment in diverse regions, including India, where government initiatives coexist with significant operational hurdles [15].
Telemedicine's expansion post-COVID-19 has extended across multiple special-ties worldwide. For instance, Latin America has seen telehealth innovations improve rheumatology care by enhancing patient monitoring and access [16]. Similarly, virtual respiratory care exemplifies adaptive strategies that maintain service continuity during pandemic disruptions [6]. However, despite widespread adoption, concerns about exacerbating healthcare disparities persist. Inequities in access related to socioeconomic status, language barriers, and marginalized populations — such as transgender individuals — demand culturally competent and inclusive telehealth frameworks [17,18,19,20].
Within this broad context, obstetrics and gynecology (OB/GYN) represent a particularly vital domain for telemedicine integration. Prenatal and postnatal care are highly dependent on timely access, monitoring, and patient adherence, all areas where telehealth has demonstrated substantial benefits [21,22]. Remote antenatal monitoring using wearable devices, virtual consultations for low-risk pregnancies, tele-ultrasound interpretation, tele-abortion services, and digital patient education platforms have been implemented to various degrees globally [21,22,23]. These interventions reduce patient burden, improve care continuity, and support early detection of complications [22,24].
Comments 2: (2) Presentation and Structure of Results
- The entire Results section, especially sections with numerous citations.
Example: `Across the U.S., India, and the Middle East, 60–90% of antenatal and postnatal visits were conducted via video or phone during COVID-19 peaks`
Main text: `Table 3. Thematic summary...`
Suggested Improvements
The Results section comprehensively presents data from the 63 reviewed papers and is extremely informative. However, some statements merely list facts, leaving the reader to understand the relationships between themes and interpret them comprehensively. Furthermore, the style of citing more than 10 references for a single statement, while ensuring accuracy, makes it difficult to track which references support which values. Rather than simply listing numbers, the implications and trends they suggest should be supported with text. For example, `3.2. Utilization patterns and adoption rates` and `3.8. COVID-19: Catalyst for structural shift` are closely related. By integrating these and describing them as a single story of "an explosive increase in utilization triggered by COVID-19 and its subsequent establishment," it becomes possible to present more analytical results.
Suggested revision: "The use of teleOB/GYN services surged by over 500% in the early stages of the pandemic [4,11,16,30]. Since then, it has become established as roughly 10% of all services in high-income countries, and hybrid care is becoming a new standard."
Table 3 comprehensively maps each topic to citations, so please consider citing only key references and representative studies in your text. This will improve the readability of your text.
Suggested revision: "For example, one study in the United States reported a sharp increase in usage from less than 1% pre-pandemic to 17% [11], and in India, more than 80% of obstetricians and gynecologists have switched to telecare [3,6]."
These revisions will elevate the Results section from a simple list of data to a more thought-provoking piece that incorporates the author's analytical perspective.
Response 2: [The entire section 3. Results is restructured.
(2) Subsections: `3.2. Utilization patterns and adoption rates` and `3.8. COVID-19: Catalyst for structural shift` are integrated and described as follows:
3.1. COVID-19 as a catalyst for structural transformation
The onset of COVID-19 triggered an unprecedented surge in tele-OB/GYN services. Utilization increased by over 500% during the early pandemic period [13], with U.S. OB/GYN visits conducted virtually rising from <1% to 17% [24], and 82% of Indian obstetricians adopting telecare by mid-2020 [15]. In the Middle East, widespread uptake was similarly reported [14], while digital health investment soared by 1818% to USD 788 million in Q1 2020 alone [39]. In Latin America, platforms previously absent saw rapid adoption, transforming telehealth into a structural component of care [16]. By 2022, hybrid care models were adopted by 71% of OB/GYN clinics in high-income countries [42], with routine remote consultations accounting for 9–12% of services [43], signaling a shift from emergency response to sustained delivery mode [12].
(3) The sentence is revised as follows: Utilization increased by over 500% during the early pandemic period [13], with U.S. OB/GYN visits conducted virtually rising from <1% to 17% [24], and 82% of Indian obstetricians adopting telecare by mid-2020 [15].
(4) Table 3: Only key references and representative studies are considered in citation.
]
Thank you for pointing this out. We agree with this comment. Therefore, we have restructured the entire section 3. Results
“[updated text in the manuscript and hight lighted]”
Comments 3:
(3) Connection between the critical points and the results in the Discussion section
Points of criticism
Text: `4. Discussion`
`Many existing analyses are biased toward telemedicine success stories, often showcasing uptake metrics... However, these portrayals omit valuable counter-narratives of disruption and clinical compromise. `
`A major gap in the current evidence base is the limited representation of low-resource and conflict-affected contexts...`
`A core shortcoming of the reviewed literature is the absence of critical scrutiny regarding unsuccessful implementations, adverse outcomes, or discontinued telemedicine programs. ` (lines 364-366)
Points of criticism and suggested improvements
The Discussion section is one of the strongest points of this paper, as it incisively points out biases and gaps present throughout the literature. However, the connection between these critical points and the analysis of the 63 reviewed articles is somewhat unclear. Readers cannot determine whether these gaps were "not included at all in the 63 studies reviewed" or "spotlighted in a few studies but ignored overall."
[Suggested Improvement]
More explicitly link each point in your discussion to the review results (or lack thereof).
"Of the 63 studies we reviewed, none focused primarily on the failure or discontinuation of telemedicine implementation, or associated adverse events. While a few studies mentioned diagnostic delays [58,63] and diagnostic discordance [57], these were reported only as secondary findings in the context of success stories. This suggests a strong publication bias in this field and a systematic loss of opportunities to learn from failures."
By providing evidence based on your own review results in this way, your argument will be more persuasive and rooted in the analysis of the studies, rather than simply a general critique.
Our discussion underscores the need for rigorous methodological frameworks... This aligns with Qian et al. [64]... Similarly, Liu et al. [65]...` This paragraph contains detailed citations and comparisons of studies not included in the review (e.g., heart sound analysis, home nutritional management).
Citing studies not included in the review [64, 65] to support the argument for methodological rigor is an original approach. However, as a scoping review, this may seem abrupt and obscure the focus of the discussion. The main purpose of the discussion is to synthesize and interpret the findings from the literature reviewed.
[Suggested Improvement]
The argument made in this paragraph, "the need for more rigorous methodology (advanced AI assessment, longitudinal follow-up)," has already been thoroughly discussed in the preceding discussion. Therefore, deleting this paragraph or significantly simplifying it would likely improve the consistency of the discussion. If this comparison is retained, it should be made more general rather than introducing specific studies in detail.
"For teleOB/GYN research to mature, more rigorous study designs incorporating cutting-edge AI model evaluation methods and longitudinal outcome assessments over several years, similar to those seen in other fields such as cardiology and oncology, are essential."
This will allow the authors to effectively communicate their arguments without overly relying on external examples.
Response:
Response 1: [The entire section 4. discussion is restructured. Studies Qian et al. [64] and Liu et al. [65] were deleted.] Thank you for pointing this out. We agree with this comment. Therefore, we have re-write and restructured The entire section 4. Discussion which high lighted in the manuscript.
Submission Date
01 July 2025
